# Neurovascular Manifestations of Iron-Deficient Anemia: Narrative Review and Practical Reflections through a Teaching Case

**DOI:** 10.3390/jcm11206088

**Published:** 2022-10-15

**Authors:** Marialuisa Zedde, Giacomo Portaro, Laura Ferri, Francesco Cavallieri, Manuela Napoli, Claudio Moratti, Fabrizio Piazza, Franco Valzania, Rosario Pascarella

**Affiliations:** 1Neurology Unit, Neuromotor and Rehabilitation Department, Azienda USL-IRCCS di Reggio Emilia, 42123 Reggio Emilia, Italy; 2Department of Neuroscience, Imaging and Clinical Sciences, University G. d’Annunzio of Chieti-Pescara, 66100 Chieti, Italy; 3Neuroradiology Unit, Radiology Department, Azienda USL-IRCCS di Reggio Emilia, 42123 Reggio Emilia, Italy; 4CAA and AD Translational Research and Biomarkers Laboratory, School of Medicine and Surgery, University of Milano—Bicocca, Via Cadore 48, 20900 Monza, Italy

**Keywords:** iron-deficient anemia, IDA, stroke, cerebral venous thrombosis, brain MRI, aortic thrombosis, pulmonary embolism, embolic pattern

## Abstract

Anemia is one of the most frequent diseases worldwide, affecting one-third of the general population. Anemia in general and in particular, iron-deficient anemia (IDA), has been associated to a higher risk of thrombotic manifestations, including ischemic stroke and cerebral venous thrombosis (CVT), as well as systemic extra-cerebral arterial and venous thrombosis. Despite these data, anemia is seldom considered as an etiological factor of stroke. An individual case encompassing all known neurovascular and systemic arterial and venous thrombotic manifestations related to IDA is presented with the focus on clinical reasoning issues in the diagnostic pathways, starting from the neuroradiological signs. The main questions have been identified and addressed in a narrative review of the most relevant data in the literature from a pragmatic and clinical viewpoint. The presented case concerns a 46-year-old man admitted to the Stroke Unit because of acute ischemic stroke with multiple thrombi in large intracranial and extracranial vessels, multifocal ischemic lesions in several arterial territories and the concurrent finding of asymptomatic CVT, pulmonary embolism with lung infarction and aortic thrombosis. An extended diagnostic work-up excluded the main etiologies (arterial dissection, cardiac embolism, genetic and acquired prothrombotic disorders, such as cancer and antiphospholipid syndrome), except for a severe IDA, such as to require blood transfusions followed by anticoagulant therapy for the several thrombotic manifestations. Neuroimaging and systemic vascular findings have been analyzed, and the main issues proposed by the case in the diagnostic pathway have been identified and discussed in a pragmatic clinical road map reviewing the data provided by the literature. Conclusions: IDA is a common but treatable condition that, independently or synergically, may increase the risk of thrombotic events. The diagnostic and therapeutic approach has not yet been defined, and each case should be individually addressed in a pragmatic clinical road map.

## 1. Introduction

Acute ischemic stroke with neuroimaging patterns suggesting an embolic source is a not rare occurrence in both young and old patients. Usually, the underlying embolic mechanism is inferred because of the presence of multiple ischemic lesions in more than one arterial vascular territory, but it is very rare to find a thrombus in a large vessel like a “smoking gun” in the diagnostic pathway looking for an etiology of the stroke. It is even rarer to find the simultaneous presence of thrombi in multiple arteries and the association of arterial and venous thrombosis as well as cerebral and extra-cerebral thrombosis. The current etiological classifications of ischemic stroke do not include such cases in any category, and the diagnostic pathway is not clearly defined, shared and codified. Similarly, the most appropriate therapeutic approach to these situations has not been defined, and each case has an individual management. Among the different possible etiologies are acquired or hereditary prothrombotic conditions as well as common diseases that have a rare presentation in which stroke is the first manifestation; not infrequently, the diagnostic work-up does not lead to the identification of a certain proof, allowing to collect only a series of faint clues.

## 2. Case Presentation

A 46-year-old man was referred to the Emergency Department (ED) in the afternoon because of unsteadiness, vertigo and nausea, reporting the awareness of the symptoms at awakening, around ten hours before the consultation; his last time well was almost 24 h before. His past medical history was notable for two episodes of head concussion without clinical consequences and brain lesions on CT imaging, the recent finding of increased value of glycated hemoglobin and a long-lasting IDA, erratically treated with iron oral supplementation. The first blood laboratory assessment in ED confirmed the microcytic anemia without other relevant findings: hemoglobin 8.3 g/dL (normal range 14.0–17.5 g/dL), RBCs 4.26 × 106/µL (normal range 4.50–6.00 millions/µL), hematocrit 28.6% (normal range 40.0–52.0%), MCV 67.3 fL (normal range 80.0–95.0 fL), MCH 19.5 pg (normal range 26.0–32.0 pg), MCHC 28.9 g/dL (normal range 32.5–36.0 g/dL), platelets 211 × 1000 µL (normal range 150–400 × 1000/µL), fibrinogen 188 mg/dL (normal range 181.0–384.0 mg/dL), D-dimer 2485 ng/mL (normal range 10–500 ng/mL), regular liver and kidney functions, absence of electrolytic disturbances and normal inflammatory markers. The initial neurological examination was remarkable for impaired alertness (psychomotor slowness, tendency to fall asleep), dysarthria and left limbs ataxia with a global National Institute of Health Stroke Scale (NIHSS) score of 11. The patient underwent an unenhanced brain CT scan, whose findings were significant for multiple subcortical hypodensities in both cerebellar hemispheres and vermis, which was highly suggestive of subacute ischemic lesions in the territory of bilateral posterior inferior cerebellar artery (PICA) and left superior cerebellar artery (SCA) (Figure 1).

Therefore, an arch-to-vertex CT-angiography (CTA) was performed, and it revealed the following:A rounded hypodense lesion partially adherent to the aortic arch wall and partially floating within the lumen of the aortic arch, in the segment between the origin of the left common carotid artery (CCA) and the left subclavian artery (SA);A similar smaller hypodense rounded structure partially adherent to the posterior wall of the brachiocephalic artery (BCA);A progressive reduction in intensity of contrast filling of the right vertebral artery (VA) starting from the V2–V3 transition, leading to a complete occlusion in the V4 segment;A lack of contrast filling of the right lateral dural venous system involving transverse sinus (TS) and sigmoid sinus (SS) and also reaching the jugular bulb and the proximal extracranial segment of the right internal jugular vein (IJV).

The intraluminal arterial hypodensities described in (a,b) (Figure 2) are suggestive for intra-arterial partially floating thrombi with V4 VA occlusion (c) (Figure 3), with a presumed mechanism of artery-to artery embolism. Conversely, the filling defect on the intracranial venous compartment (right TS, SS and IJV) (d) (Figure 4) is highly suggestive of cerebral venous thrombosis (CVT).

The presence of multiple subacute ischemic lesions already evident in the brain CT at admission excluded intravenous thrombolysis and endovascular treatment. Therefore, the patient was admitted to the Stroke Unit for acute stroke care and further investigations and started single antiplatelet treatment and low-molecular weighted heparin at a prophylactic dose. At the admission, the patient shown a stepwise neurological worsening, and the final clinical examination at 12 h from the admission was remarkable for a right facial–brachial hemiplegia, cerebellar dysarthria, dysphagia, inconstant limitation on left lateral gaze without nystagmus and a left cerebellar syndrome. A brain MRI was performed at about 15 h from the admission, confirming the presence of multiple subacute ischemic areas, located in the left parasagittal vermis and in the medial–superior region of both cerebellar hemispheres, and outlining other subacute infarctions involving the right frontal and parietal lobe (mainly in cortical locations) and the left paramedian medullary region (Figure 5). Moreover, MR angiography (MRA) highlighted the lack of visualization of the right V4 VA and the corresponding PICA, as well as the CVT involving the right TS, SS and IJV. The number and distribution of the subacute ischemic lesions in several vascular territories is coherent with the location of thrombi in large arteries as showed by CTA, i.e., right VA, BCA with potential involvement of the ipsilateral ICA and VA territories, and in the aortic arch near the origin of the left SA.

Further tests were carried out trying to identify other prothrombotic factors and to clarify the etiology of the clinical picture. The extended laboratory investigations included normal total (360.6) and unsaturated iron-binding capacity (283.6), transferrin saturation (21.3%), serum ferritin (39 ng/mL, normal range 22–275 ng/mL), serum iron (77 µg/dL, normal range 60–170 µg/dL) and transferrin (284 mg/dL). The fecal occult blood test was negative; nocturnal paroxysmal hemoglobinuria and hemolytic anemia were excluded, and the analysis of hemoglobin subgroups resulted within a normal profile. Antibodies anti-double stranded DNA, anti-mitochondrial (AMA), anti-smooth muscle (ASMA), anti-nuclear (ANA), anti-phospholipid, anti-deamidated gliadin peptide and anti-transglutaminase were within the normal range; the main onco-markers (AFP, CEA, CYFRA 21.1, NSE, PSA) did not express any significant pathological increase. Anti-thrombin levels were within the normal range, the search for a mutation of Factor V-R506Q was negative; a heterozygote genotype was identified for MTHFR-C677T, and an insignificant increase in homocysteine values was found. Transthoracic and transesophageal echocardiography as well as 24 h ECG monitoring were unremarkable. Conversely, a thoracic-abdominal CT scan highlighted thromboembolisms involving also the sub-segmental branches of both pulmonary inferior lobes and an 8 mm free thrombus adherent to a crescent-shaped atheroma of aorto-iliac junction and floating within the lumen (Figure 6).

At 30 days from the symptoms’ onset, the neurological condition was substantially unchanged. A follow-up brain MRI highlighted a regular evolution of the ischemic lesions and confirmed the persistent occlusion of the right V4 VA. Moreover, the brain MRI showed as unchanged the white matter hyperintensities in the external capsula and centrum semiovale (Figure 7) together with few subcortical punctate cavitating lesions in the posterior thalamus on both sides and a left temporal hyperintense lesion involving the cerebral cortex. All these lesions are quite suggestive of a vascular ischemic nature.

Conversely, the right TS, SS and IJV CVT was recanalized in the follow-up MRA (Figure 8), suggesting a recent timing of the intraluminal thrombus, although it was most likely an incidental finding, and the partially floating thrombus in the abdominal aorta also disappeared in the follow-up CTA (Figure 8).

Finally, a whole-body G18-FDG PET did not show abnormal metabolic activities except for both basal pulmonary areas and for the gastro-duodenal region. The former one was explained by the pleural effusion secondary to pulmonary embolism, while the latter was explored with endoscopic and histologic studies that did not reveal abnormalities.

During the hospital stay, the neurological condition was substantially unchanged. The severity of anemia in the acute phase requested a blood transfusion (two units of packed red blood cells), which was followed by iron intravenous infusion and vitamin replacement. The hemoglobin level progressively increased until a persistent stabilization above 12 g/dL. LMWH dose was increased to an anticoagulant posology after blood transfusion and insulin therapy was set up.

Even though a large panel of investigations was performed, we were not able to identify other genetic or acquired prothrombotic factors but IDA nor a cause of bleeding that could explain the microcytic anemia. In consideration of the multi-district thrombosis, involving also the pulmonary district, after the hospital discharge, oral anticoagulation with rivaroxaban 20 mg was started for venous thromboembolism (VTE) treatment. A 12-month follow-up did not raise any other etiology.

## 3. Discussion

This clinical case raises several questions both in the diagnostic steps and in the therapeutic choices and has almost unique characteristics, joining together in the same patient simultaneous arterial and venous thromboses in multiple sites. Starting from the symptoms, which led the patient to the ED, the first neuroimaging studies in ED made already evident the clinical complexity related to the presence of multiple sites of thrombotic occlusions in the large arteries supplying the brain and in the cerebral venous sinuses. While arterial thrombosis was symptomatic and its acute timing was easily defined by the presence of a recent multifocal ischemic stroke, CVT was an incidental and asymptomatic finding and, as such, undated. Only later, the resolution of CVT with anticoagulation did support its recent timing. At this step, even with a history of chronic IDA, the causal hypotheses for simultaneous arterial and venous thrombosis were substantially limited to genetic or acquired prothrombotic conditions, which is in accordance with the existing stroke classifications. A genetic thrombophilia would have had a lower probability because of the simultaneity of thrombotic findings and the absence of a previous history of thrombotic events, even minor ones. Moreover, the simultaneous involvement of the venous and arterial sides virtually excluded major cardioembolic sources as atrial fibrillation (AF), and a paradoxical embolism mechanism would not be able to take account of the etiology of venous thrombi. However, the patient did not have a defect of the interatrial septum, and two weeks of heart monitoring did not identify atrial fibrillation.

As a general construct, the presence of multiple cerebral ischemic lesions with an embolic pattern and without an identifiable known etiology leads to the consideration of the embolic stroke of undetermined source (ESUS) [1] concept. From its proposal, mainly as a tool to select patients for trials of antithrombotic treatment, it underwent many refinements with recent reappraisal in a more complex and subcategorized form [2]. A recent well-structured criticism of the ESUS concept, as proposed until now, has been published [3], starting from the too optimistic assumption that the majority of ESUS were thromboembolic and therefore probably treatable with anticoagulants. Another issue is that the proposed minimum diagnostic pathway to define a stroke as ESUS is not able to answer to the multiplicity of etiological phenotypes included the main neuroimaging pattern of ESUS. Moreover, several concurrent etiologies may be simultaneously present in an individual patient with stroke.

In the diagnostic work-up of cryptogenic embolism, the co-occurrence of venous and arterial thrombi in cerebral circulation was suggestive for two main diagnostic hypotheses, i.e., antiphospholipid antibodies syndrome (APS) and cancer-associated stroke (CAS). APS is a disease classically associated to many thrombotic manifestations in arterial and venous districts [4,5,6], but it was excluded by the normal findings of the autoimmunity blood tests. Moreover, the neuroimaging features of APS-associated stroke are not completely congruent with the presented case. Indeed, an interesting study [7] comparing APS-associated stroke and AF-associated stroke starting from cerebral cryptogenic embolism and in particular multi-territory lesions showed that the first one has mild neuroimaging features (small lesion prevalence, smaller infarct volume, and absence of relevant artery occlusion).

CAS was a meaningful diagnostic hypothesis, and several issues may raise it in the context of the diagnostic work-up of cryptogenic embolism [8]. Indeed, in clinical neurovascular practice, the main acquired cause of cerebral and systemic coagulopathy is cancer. Stroke has been reported as the first manifestation of an occult malignancy in up to 3% of patients [9,10]. Approximately, up to 20% of the patients with cryptogenic stroke have an underlying unknown malignancy [11], and stroke as a complication of known cancer increases the morbidity and mortality. The burden of cerebrovascular diseases in cancer patients is not negligible, because it is the second most common neurological manifestation following metastases [12] with a 6-month incidence of 3.0% vs. compared with 1.6% in control patients (HR: 1.9; 95% CI: 1.8 to 2.0) [13]. Among patients with cryptogenic cerebral embolism, CAS with a first diagnosis of previously unknown cancer is about 20%, and the main features are higher D-dimer levels (over 20 times higher than those without cancer are) and a stroke pattern on neuroimaging with multiple lesions in multiple vascular territories [14]. These features, proposed again and confirmed in other larger studies, have been proposed as useful clinical clues to select patients to screen for hidden malignancy. The first issue, i.e., the neuroimaging pattern of stroke with multiple acute cerebral infarcts on DWI-MRI, has been proposed as highly specific of CAS [15]. This neuroimaging finding has been so much emphasized as an MRI marker of CAS that it produced a dedicated sign, the Three Territory Sign (TTS), which was proposed as a highly specific marker and is six times more frequently observed in CAS than AF-related ischemic stroke; thus, for patients with TTS, the screening for an underlying malignancy is suggested [16].

Moreover, a multiple scattered lesion pattern, often reported in CAS, was associated to higher D-dimer values [17,18,19]. These findings support the link between infarction in multiple vascular territories and cancer-associated hypercoagulation as the underlying stroke mechanism.

Unfortunately, these clues have some limitations. In particular, because stroke itself is frequently associated to an increased D-dimer, its value alone is not sufficient for this purpose. D-dimer is a more general marker of an activated coagulation system, and the presence of thrombi and ischemic lesions in the presented case is usually associated to the finding of increased D-dimer values independently from the cause of the thrombosis. Another surrogate marker of the pro-thrombotic role of cancer is the detection of High-Intensity Transient Signals (HITS), also known as microembolic signals, on Transcranial Doppler (TCD) [20], but in the present case, the need of this detection as a diagnostic clue is largely overcome by the identification of thrombi in several large vessels.

The finding of multiple thrombi in large vessels, both arteries and veins, in the presented case deserves a dedicated consideration as a potential clue for a hidden malignancy, since hypercoagulability is the most relevant mechanism of CAS [21] and known in its more catastrophic appearance as Trousseau syndrome. It has been associated to the occurrence of multiple territories ischemic stroke [22]. It has been demonstrated in an interesting study [23] that patients with CAS with embolic pattern have an elevated risk of associated VTE and arterial thromboembolism with a negative impact on the 1-year prognosis. In this study, VTE has been considered also as deep vein thrombosis in legs and pulmonary embolism, but other unusual locations of venous thrombosis have not been assessed. Moreover, recurrent venous and arterial thromboembolic events are often associated in patients with active cancer despite anticoagulation [24].

If the diagnostic pathway of the presented case was analyzed according to the advanced diagnostic protocol (Figure 9) proposed in cryptogenic embolism [8], the results would be unsatisfactory, and a univocal characterization would not be reached (Figure 10).

Indeed, the neuroimaging pattern of the patient does not fulfill a single category, but it combines features belonging to the three DWI patterns and leaves open the possibility of multiple etiologies except for branch occlusive disease. The multiplicity of lesions (with an embolic pattern and in small vessels territory) and the recent and past timing of the findings is hard to lead back to a single category. In the presented case, a hidden malignancy was extensively looked for, but all studies were negative in this regard both in the acute phase and in the 12-month follow-up. Other, even rare, recognized causes of stroke and thrombosis have been investigated and excluded. At this point, IDA was considered with more attention, focusing on the described mechanisms linking IDA and thrombosis and on the literature data.

Anemia is one of the most frequent diseases worldwide and, although the great differences due to age, sex and geographical distribution make it difficult to assess the real incidence, it roughly affects one-third of the general population [25,26,27]. Among all the causes, IDA is considered the most frequent, and poor dietary intake, blood loss, reduced iron absorption or chronic diseases [27] can sustain it. Identification of the underlying causes and iron supplementation represent the mainstay of the management. In the last few years, several studies raised the association between anemia, especially IDA, and an increased thrombotic risk. Indeed, anemia has emerged as a risk factor for cerebrovascular diseases, and many types of evidence have demonstrated how the recognition and treatment of underlying anemia might improve the outcome in patients with acute stroke, although its role as an independent risk factor is still controversial [27]. In this regard, a few years ago, Maguire et al. [28] outlined how, in a population of children without previous relevant diseases, IDA was more frequent in patients who developed stroke than in controls. They also acknowledged IDA as a consistent vascular risk factor, given that it affected around half of the children presenting with stroke without other possible etiologic factors. Moreover, they also stated that CVT were even more frequent than ischemic stroke in children with IDA. In adult patients, the early reports of arterial thrombosis associated with IDA were anecdotal. Yakushiji [29] described two cases of ischemic stroke from embolic sources in female patients with aortic floating thrombi without an atheromatous ground and with IDA as the main risk factor. Earlier, Akins et al. [30] collected three cases of patients with transient focal neurological symptoms or ischemic stroke associated with carotid artery thrombosis. They had in common the absence of any of the classical prothrombotic risk factors and the finding of IDA, which was also complicated by thrombocytosis. On the other hand, the venous district may also suffer from the consequences produced by anemia. Indeed, in the same way as in children, IDA has been accounted among the risk factors for CVT. A recent case-control study [31] confirmed a higher prevalence of CVT in patients with anemia, especially in microcytic forms, and outlined an inverse correlation between the value of hemoglobin and the risk of developing thrombosis in the venous compartment. In particular, the association of IDE with reactive thrombocytosis has been considered as the main factor responsible for the hypercoagulable state underlying the increased risk of VTE as a component of Virchow’s triad. In a population-based case-control study in an Asian population [32], the association between IDA and VTE was statistically significant (3.41 vs. 2.06%, respectively, *p* < 0.001) and the odds ratio (OR) of previous IDA for subjects with a VTE was 1.43 (95% confidence interval (CI): 1.10–1.87) compared with the controls. The mechanisms of the association between IDA and increased thrombotic risk, both arterial and venous, however, are not fully clarified, and the simultaneous finding of multiple arterial and venous thrombotic events in the same individual has been reported only in an anecdotal way and often evokes etiologies different from anemia in the differential diagnostic pathway. Moreover, the prothrombotic role of IDA therefore appears to be considerably underestimated in neurovascular clinical practice, and the diagnostic and therapeutic pathways of these patients have yet to be defined, raising still unanswered questions.

The association between IDA and thrombotic risk is further increased by thrombocytosis. To this regard, the development of an elevated platelets count, probably due to the hyperactivity of erythropoietin in anemic state, has been considered one of the potential thrombotic mechanisms [33]. Indeed, some studies described a higher number of thrombotic events in IDA associated with thrombocytosis than in patients with a normal platelets count [31]. Analyzing other cases of stroke associated with IDA, other hypotheses have been postulated about their connection. First of all, the reduction in hemoglobin levels may produce a decrease in oxygen concentration in the blood flow, and the brain territories could be the most affected by this change [32,33]. Moreover, a possible endothelial dysfunction has been taken into account, considering that it may be the result of a hyperkinetic condition of the blood flow as well as the steady inflammatory state, which are both produced by persistent anemia [27]. All these factors, together with a possible dysfunction of erythrocyte kinetic abilities, may increase the risk of infarction in watershed territories [27].

A further consideration relating to this and other attempts at the phenotyping of cryptogenic embolism, both before and after the introduction of the ESUS concept, is that none of these includes the finding of thrombi in large vessels as a distinctive element nor the coexistence of arterial and venous thrombosis or the simultaneous presence of cerebral and systemic thrombosis. In some of these classifications, the cerebral and systemic ischemic lesions are considered, but the identification of thrombi in the vessels is not among the classifiable items. Interestingly, also, the more detailed subtyping proposed by the critics of ESUS concept [3] included as a separate category cancer-associated coagulopathy but not specifically anemia as a known prothrombotic factor.

To our knowledge, there are no previous case descriptions of simultaneous arterial and venous thrombosis in patients with IDA as isolated prothrombotic factors. However, if the prothrombotic power of anemia has been widely accepted, how and if it may independently be responsible for arterial and/or venous thrombosis has not been completely clarified. Nevertheless, our case confirms how IDA might be considered, de facto, a strong prothrombotic factor, and it is able to produce a multi-district systemic thrombosis in a young patient without other relevant medical conditions. Thus, it should be important to point attention to a treatable disease, such as IDA, and its strict association with a life-threatening condition, such as cerebral and systemic thrombosis. Moreover, if we consider that vascular events may be related to hardly modifiable risk factors, anemia falls into the category of treatable conditions, and its correction may affect the outcome of many patients. A key issue might be that despite or due to its high incidence in the general population, anemia could be underdiagnosed. Furthermore, a poor compliance to long-term iron replacement could influence the success of the therapy, as happened to our patient. Surely, raising the awareness on the importance of an earlier identification of anemia and its causes might be a critical target, although it was not been demonstrated that treating anemia reduces the risk of thrombotic events.

## 4. Conclusions

This unique clinical case raised attention regarding a common but treatable condition that, independently or synergically, may increase the risk of thrombotic events in all body districts and across all ages. IDA should be considered as a cause of ischemic stroke, mainly if it is associated with the evidence of thrombotic occlusion of large vessels and with systemic and venous thrombosis.

## Figures and Tables

**Figure 1 jcm-11-06088-f001:**
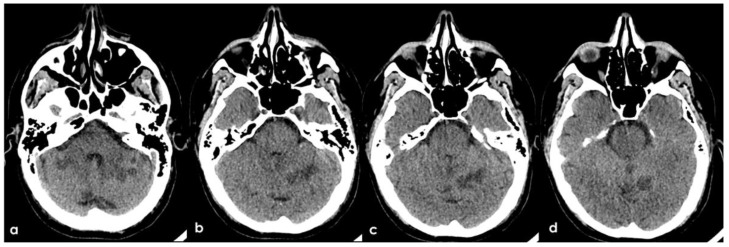
Brain CT performed at the admittance to the ED. From a to d: ascending axial CT slices of the posterior cranial fossa, showing bilateral subcortical multiple rounded hypodensities in the cerebellar white matter (**a**) in PICA territory with left side prevalence (**b**,**c**) and a similar hyperintense lesion in the superior part of vermis (**b**–**d**) and in the left anterior lobe of the cerebellum, supplied by the superior cerebellar artery (SCA).

**Figure 2 jcm-11-06088-f002:**
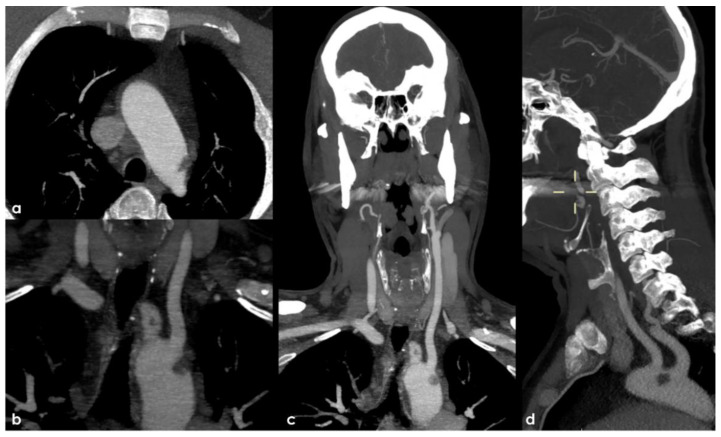
Arch-to-vertex CTA. The main findings are illustrated in the pictures from a to d: (**a**) Axial source slice at the aortic arch level with an irregularly rounded hypodense lesion adherent to the posterior–superior wall of the aorta and partially floating into the lumen; (**b**) Maximum Intensity Projection (MIP) reconstruction on a coronal plane at the level of the hypodensity seen in a and showing the spatial relationship with the left CCA origin (magnified picture). A similar smaller rounded hypodensity, apparently floating into the BCA lumen is also evident; (**c**) The same MIP coronal plane as in b with a minor magnification, allowing to better appreciate the final potential locations of artery-to-artery embolism from the above signaled thrombotic formations; (**d**) MIP reconstruction in a sagittal plane showing the site of the aortic arch thrombus between left CCA and left SA.

**Figure 3 jcm-11-06088-f003:**
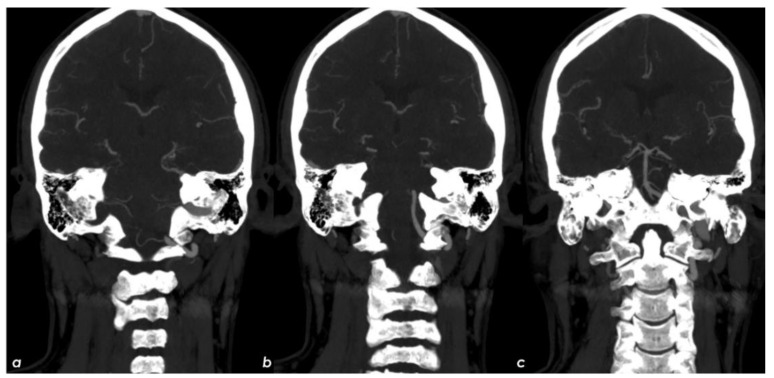
Arch-to-vertex CTA with MIP reconstructions in the coronal plane. The lack of contrast filling of the right distal V3 (**a**) and V4 (**b**) VA is evident in comparison to the left VA starting from the dural ring with retrograde filling of the pre-junctional segment (**c**).

**Figure 4 jcm-11-06088-f004:**
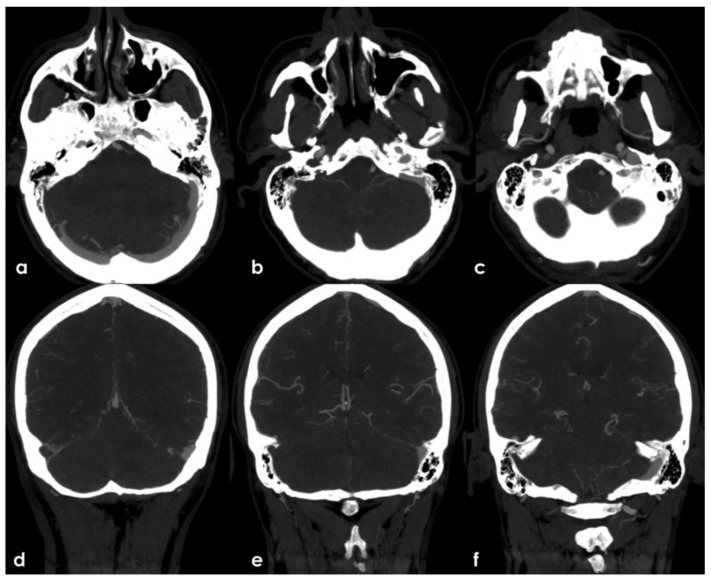
Arch-to-vertex CTA with MIP reconstructions. The lack of contrast filling of the right TS (from its middle segment), SS and IJV in comparison with the regularly contrast filled contralateral segments is showed in axial cranio-caudal slices (**a**–**c**) and in coronal posterior-to-anterior slices (**d**–**f**).

**Figure 5 jcm-11-06088-f005:**
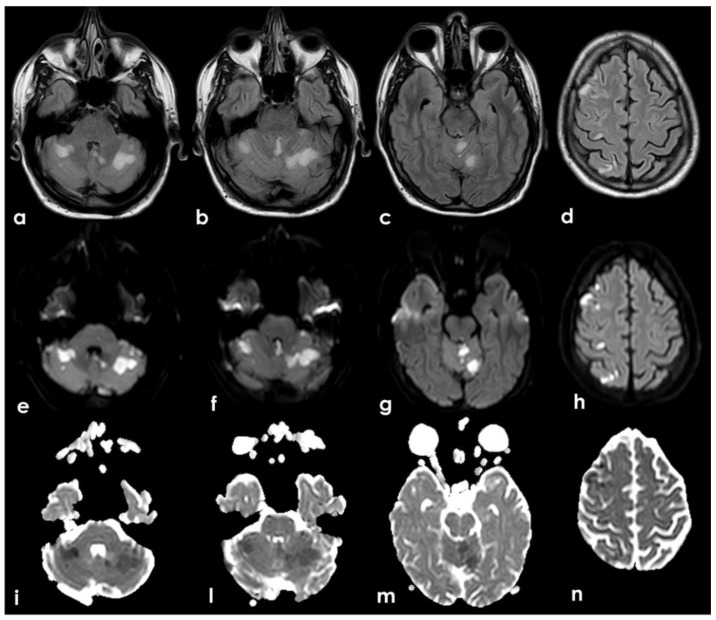
Brain MRI. Axial Fluid Attenuated Inversion recovery sequences (**a**–**d**) and the corresponding Diffusion Weighted Imaging (DWI) (**e**–**h**) and Apparent Diffusion Coefficient (ADC) (**i**,**l**,**m**,**n**) images showing multiple recent ischemic lesions in cerebellar hemispheric white matter and vermis and in the frontal and parietal right cortex with involvement of the cortical–subcortical junction. All the areas of signal change are hyperintense on FLAIR and DWI sequences and hypointense on ADC, and these features are coherent with the ischemic nature and the subacute timing.

**Figure 6 jcm-11-06088-f006:**
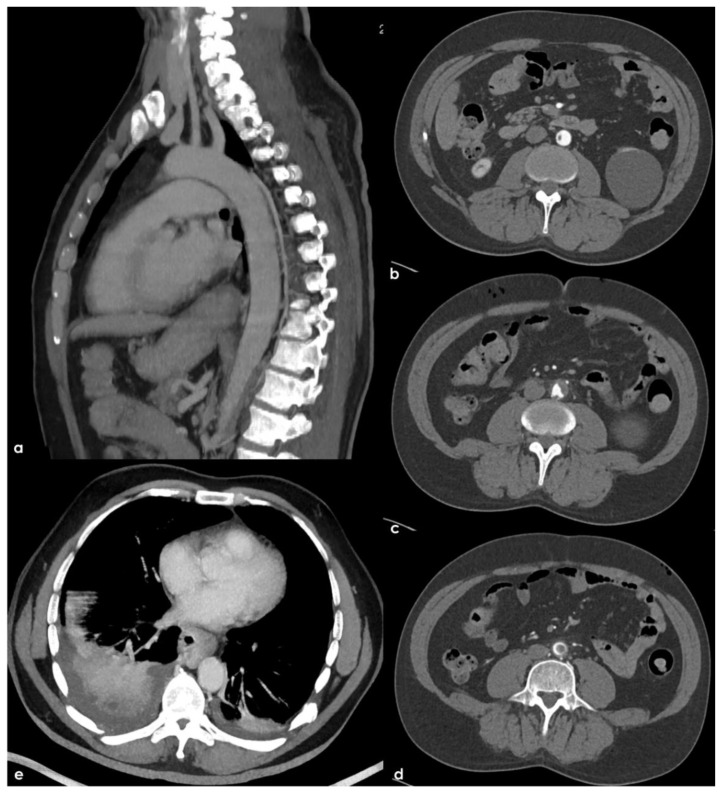
Thorax and abdomen CTA with Maximum Intensity Projection (MIP) reconstructions. In (**a**), a sagittal plane view of the left half of the aortic arch and descending aorta demonstrated the disappearance of the thrombotic hypodensities previously seen (Figure 2). Consecutive axial slices of the pre-terminal segment of the abdominal aorta ((**b**–**d**), rom cranial to caudal tip) showed a hypodense structure arising from a crescent-shaped structure in the aortic wall with atheromatous features and partially floating into the aortic lumen. The occlusion of sub-segmental branches of the pulmonary artery on both sides with a lung infarction on the right side is illustrated in (**e**).

**Figure 7 jcm-11-06088-f007:**
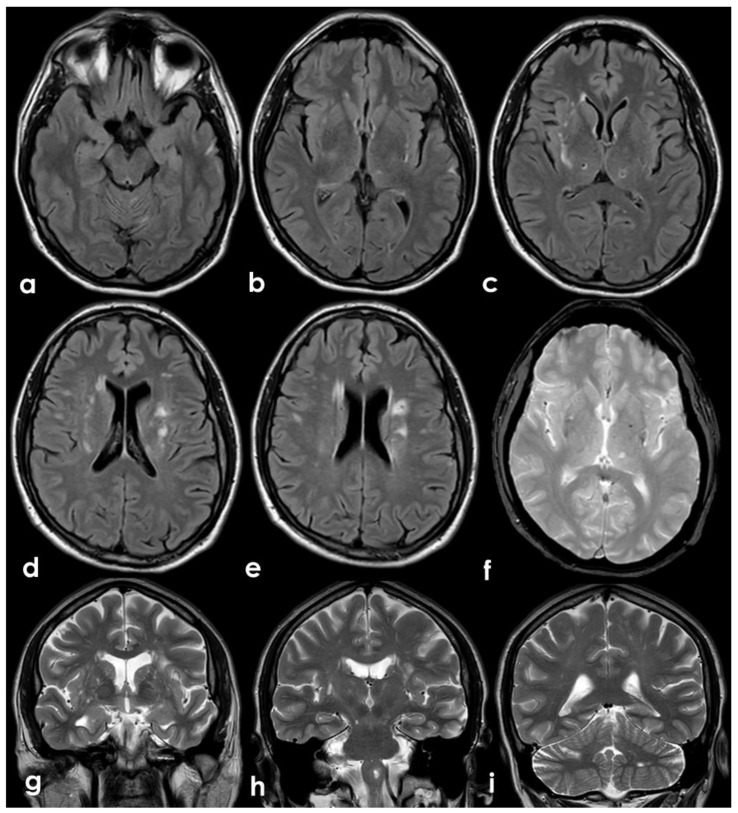
Brain MRI performed after 4 weeks from the admission. Axial FLAIR images (**a**–**e**) showing multiple white matter hyperintensities in the deep and subcortical location, (i.e., centrum semiovale, external capsula) and a few cavitating lesions in the posterior thalamus on both sides (**b**,**c**). In (**a**,**b**), a focal cortical hyperintense lesion is identified in the left temporal lobe. Gradient Echo (GRE) sequences (**f**) show a single deep small and rounded hypointense area (microbleed), and the T2-weighted sequence images in the coronal plane (**g**–**i**) show several enlarged perivascular spaces, mainly in basal ganglia on both sides.

**Figure 8 jcm-11-06088-f008:**
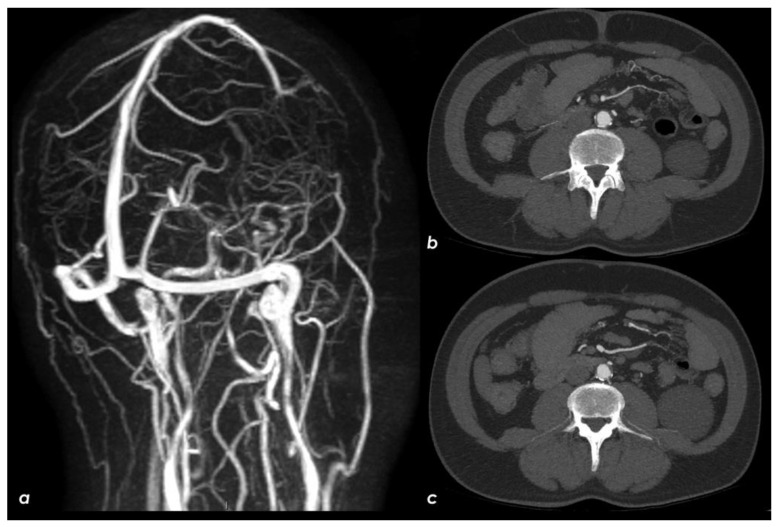
Brain MRI and abdominal CTA in follow-up. The right lateral dural venous system was patent in the MRA ((**a**), time of flight reconstruction). Aortic CTA showed the disappearance of the floating hypodense structure previously seen and adherent to a crescent-shaped atheroma ((**b**,**c**), axial MIP reconstructions).

**Figure 9 jcm-11-06088-f009:**
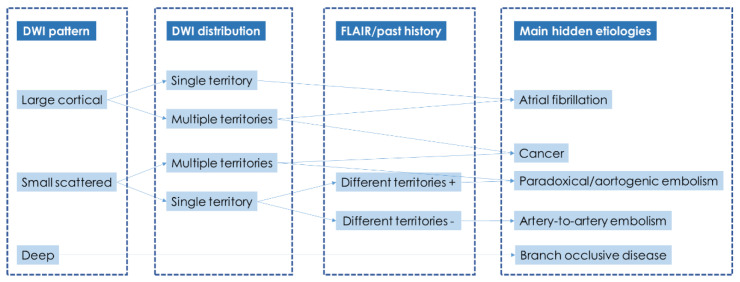
DWI/FLAIR MRI findings as tool for hypothesizing the main hidden etiologic categories in cryptogenic embolism (simplified and adapted from Bang OY) [16]. The categorization is based on the step-by-step analysis of: Diffusion Weighted Imaging (DWI) infarct pattern: embolic versus deep and large versus small scattered; DWI infarct distribution: ≥1 vascular territory involved; Past stroke on history or fluid-attenuated inversion recovery (FLAIR) image: the same side versus different territory.

**Figure 10 jcm-11-06088-f010:**
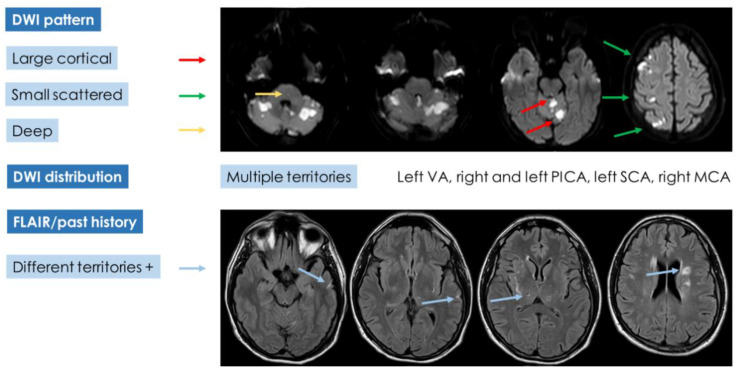
Application of the categorization proposed in Figure 9 to the presented case.

## Data Availability

Not applicable.

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
