# Peer review of "Neurovascular Manifestations of Iron-Deficient Anemia: Narrative Review and Practical Reflections through a Teaching Case"

_jcm, 2022, doi:10.3390/jcm11206088_

Round 1

Reviewer 1 Report

Nicely presented manuscript, centered on the presentation of a case of a patient who acute ischemic stroke with multiple thrombi and the neuroradiological and laboratory work up verified that it was a case of IDA related stroke.

I raise a concern about the category under which this manuscript should be joined. I consider that this manuscript is not a narrative review but rather a well presented case report with a discussion section that referrs to the underlying causes of stroke, with a special subinterest to IDA.  It is not common practice for a manuscript that is destined to to be concidered as a review article to be based on a case description. Apart from that, a manuscript that that is a narrative review centered on the neurovascular manifestations of iron-deficient anemia dedicates its main part on that subject and not on a general review regarding the pathophysiology that underlies the entity of ischemic stroke. There is an uneven distribution of the conents of your manuscript, wich is mainly centered on the presentation of a single case and a thorough review of the etiology of ischemic stroke. A minor part of your review is in accordance with the title of your manuscript, that is on iron-deficient anemia.

Author Response

First of all, we would thank the reviewer for reading and appretiating our paper. 

We understand the comment of the reviewer about the classification of the paper. We chosen to propose it as a narrative review because our aim is to use the presented case as a tool to describe the critical issues of the differential diagnosis in a complex situation, highliting the limitations of the existing classifications of stroke etiology. The title is focused more on the conclusions than on the many steps of the diagnostic process but we proposed it because of the detailed description of the case. We are hoping that the details in the description and in the neuroimaging issues and findings may help the reader to think out of the box in specific situations and to be broaden the range of the differential diagnosis in a smart approach. This is the reason why we explicited the application of the proposed classifications of cryptogenic embolism to put patient in figure 9 and 10.  

We will defer to the Editor's decision on the classification of the paper, fully understanding the comments of the reviewer. 

Reviewer 2 Report

Thank you very much for this very interesting case report and the extensive work-up you have done. If I may suggest one thing it would be to shorten it in order to make it easier to read, e.g. the section about the MRI infarct pattern including figures 9 and 10 seem seem a bit lengthy  for neuroradiology educated readers.

Author Response

We would thanks the reviewer for appreciating our paper. 

We fully understand the suggestion about shortening the description of the neuroimaging pattern according to the target readers. We are not really sure of the neuroradiological education of our potential readers because Journal of Clinical Medicine has a multisciplinary range of readers, so we chosen to make explicit some considerations in order to facilitate the understanding of the main clinical reasoning for the unexperienced reader (from the neuroimaging point of view). In a final analysis most of the presented clinical reasoning started from the neuroimaging and imaging pattern. The application of the classification of cryptogenic embolism to our case in figure 9 and 10 has the purpose to highlight the limitations of these classifications in clinical neurovascular practice and it is the natural conclusion of the criticisms about the ESUS construct. 

We fully agree on shortening this section if the expected readers are familiar with neuroimaging issues, so we will follow the indications of the editors. 

We would like to propose to shorten this section if the expected readers , if the Editor agrees,  

Reviewer 3 Report

This is an important case report on massive tromboembolism. I have only a few comments.

1. The authors give laboratory data that are not in SI units. This is OK but most clinicians in Europe may have difficulties to interpret the findings. I suggest the authors provide the normal values or normal value ranges from their laboratory. That would help.

2. The English language is rather good but might be improved. I suggest the text is subjected to a professional English language revision. Such revisions may be available in Reggio Emilia but can also be performed on the internet.

Author Response

First of all, we would thank the reviewer for the nice comments on our paper. 

We modified the draft accordingly. 

Point 1: We added in the text the values of the normal range for the laboratory data.

Point 2. We addressed the issue of improving the language with the help of a native English speaker, making simpler and less hypotactic the synthax. Reading again the draft after the comment of the reviewer together with a native English speaker allows us to identify the main sentences to change. 

Round 2

Reviewer 1 Report

I concider that this manuscript could be a successful candidate for publication, although I insist on my comment regarding the classification of the paper. I understand the concept of the authors but strict criteria should be used regarding the most proper subcategory that is relevant for any particular manuscript. These criteria should be universally accepted and my opinion is that this manuscript does not fullfill the criteria in order to be categorized as a narrative review.